# A Simulation Study on the Crack Propagation Behavior of Nanostructured Thermal Barrier Coatings with Tailored Microstructure

**Lei Zhang [1], Yu Wang [1,2,\*], Wei Fan [1,3], Yuan Gao [1], Yiwen Sun [1] and Yu Bai [1,\*]**

[1]   State Key Laboratory for Mechanical Behavior of Materials, Xi'an Jiaotong University, Xi'an 710049, China;
    lq925742369@stu.xjtu.edu.cn (L.Z.); fanwei@nuc.edu.cn (W.F.); joywolf0927@stu.xjtu.edu.cn (Y.G.);
    yongbaohu@stu.xjtu.edu.cn (Y.S.)

[2]   State Key Laboratory of Electrical Insulation and Power Equipment, Xi'an Jiaotong University,
    Xi'an 710049, China

[3]   School of Energy and Power Engineering, North University of China, Taiyuan 030051, China

\*   Correspondence: wangyu0730@xjtu.edu.cn (Y.W.); byxjtu@mail.xjtu.edu.cn (Y.B.)

**Abstract:** The initiation and propagation of cracks are crucial to the reliability and stability of thermal barrier coatings (TBCs). It is important and necessary to develop an effective method for the prediction of the crack propagation behavior of TBCs. In this study, an extended finite element model (XFEM) based on the real microstructure of nanostructured TBCs was built and employed to elucidate the correlation between the microstructure and crack propagation behavior. Results showed that the unmelted nano-particles (UNPs) that were distributed in the nanostructured coating had an obvious "capture effect" on the cracks, which means that many cracks easily accumulated in the tensile stress zone of the adjacent UNPs and a complex microcrack network formed at their periphery. Arbitrarily oriented cracks mainly propagated parallel to the x-axis at the final stage of thermal cycles and the tensile stress was the main driving force for the spallation failure of TBCs. Correspondingly, I and I–II mixed types of cracks are the major cracking patterns.

**Keywords:** yttria-stabilized zirconia; thermal barrier coatings; unmelted nano-particle content; thermal stress distribution; cracks propagation

## 1. Introduction

The energy efficiency of a gas-turbine engine is highly dependent on the operating temperature of its hot sections, which can be greatly increased by the application of thermal barrier coatings (TBCs) [1–4]. Although many candidates of TBCs have rapidly developed, such as pyrochlore-structured $La_2Zr_2O_7$, fluorite-structured $La_2Ce_2O_7$, magnetoplumbite-type $LnMgAl_{11}O_{19}$ (Ln: La, Gd, Nd et al.) and perovskite-type $BaZrO_3$ or $SrZrO_3$, yttria stabilized zirconia (YSZ) is still an irreplaceable material for TBCs due to its good mechanical and thermal properties [5–9]. In particular, the nanostructured YSZ coatings showed superior strain tolerance and thermal insulation performance owing to their typical bimodal structure, in which many unmelted nano-particles (UNPs) were randomly distributed [10,11]. The bimodal structure of nanostructured coatings contributes to approximately 2–3 times or even more of a rise in thermal cycling life compared to their traditional counterparts [12–14].

The damage of nanostructured TBCs is related to the crack initiation and propagation caused by the residual stress, which usually originates from phase transformation, thermal expansion mismatch, thermally grown oxide (TGO) and sintering of ceramic top coat (TC). Therefore, it is of great importance to understand in-depth the degradation processes and failure modes in order to optimize the microstructures and improve the performances of nanostructured TBCs. However, the traditional

experimental or finite element modeling (FEM) methods can reveal the residual stress at a random position or the possible failure positions. Extended finite element modeling (XFEM) evolved through simulation techniques and deep development of mechanical theory. In comparison to the conventional FEM, the mesh is not restricted to the geometry or physical interface in the XFEM model. It also can simulate the crack propagation with noncontinuous characteristics and trace the crack propagation without remeshing [15–18]. Therefore, XFEM is expected to provide an effective method for studying the degradation processes and failure modes of the nanostructured TBCs.

As mentioned above, the nanostructured TBCs exhibit a typical bimodal structure that consists of recrystallized regions and UNPs. A previous study suggested that the existence of UNPs was beneficial to the improvement of fracture toughness and relaxation of residual stress [19]. However, the effect of UNPs on the crack propagation of nanostructured TBCs is still lacking systematic research. Therefore, in this study, XFEM method was employed to stimulate the thermal stress and dynamic crack propagation around the UNPs. The numerical model was based on the real microstructure of nanostructured TBCs, and some important parameters, such as plastic or creep parameters in each layer of TBCs and linear TGO growth, were taken into account with the aim of providing a methodology for designing high-performance nanostructured TBCs. It is worth noting that the UNPs (unmelted nano-particles) in this paper are identified as the concentrated areas of unmelted nano-particles embedded in the crystalline regions of coatings.

## 2. Experimental Procedure

### 2.1. Materials and Plasma Spraying Process

A commercial CoNiCrAlY powder with Y—0.4, Al—8, Cr—21, Ni—32, Co—balance (wt.%) composition was used to deposit the bond coat (BC). One $ZrO_2$-8 wt.% $Y_2O_3$ powder with 10–85 μm was selected to fabricate the top coat (TC). The morphological structure details of YSZ powder are in our previous paper [19]. The TC with 200 μm thickness was sprayed by the supersonic atmospheric plasma spraying (SAPS) system. The specimen had a disc shape with a diameter of 25 mm. The specimens were stacked on the substrate alloy GH3030 (25 mm × 6 mm) on one side. The thermal cycling test was performed by a home-made burner rig apparatus (Xi'an Jiaotong University, Xi'an, China). The coating surface was heated to 1250 ± 20 °C in 40 s by the propane-oxygen gas flame. The two sides of sample were cooled by compressed air (40 L min$^{-1}$) simultaneously, and then held for 300 s. Finally, the backside (alloy substrate) was cooled rapidly to ambient temperature by compressed air. The surface (ceramic coating) and backside (alloy substrate) temperatures were measured by two infrared thermometers (MI3, Raytek, Washington, WA, USA), continuously. The thermal cycling test was maintained at ambient temperature and normal atmosphere. The detailed information of test equipment and processes can be found in our previous work [19,20]. Four different operating conditions, described as NC1–NC4, represent the coating with different unmelted nano-particle contents (UNCs) of 11%, 15%, 22% and 34%, respectively. The spray parameters and other more details can also be found in our previous work [19].

### 2.2. Finite Element Model

#### 2.2.1. Calculation Domain

The finite element modeling was performed by commercially software Abaqus (Dassault, version 6.14, Boston, MA, USA). The TBCs system consisted a of GH4169 substrate, NiCoCrAlY bond coat (BC) and YSZ top coat (TC). Based on the real structure of TBCs, the thicknesses of the layers were set as 3 mm, 0.1 mm and 0.2 mm, respectively. According to the real structures of coatings, an initial TGO layer with thickness of 1 μm was also considered at the interfaces of TC/BC. The convex or concave presents the different amplitudes and wavelengths. The model, finite element mesh and real

cross-sectional SEM image of TBCs system are given in Figure 1. In Figure 1 the structure of the UNPs, TGO and BC layer in the model is consistent with the real structure of TBCs.

Due to the high nonlinearity of the real interface morphology, including TC/TGO, TGO/BC and BC/substrate interfaces, the mesh division has a significant influence on the accuracy of stress distribution. Therefore, two higher-order shape functions were employed in this work. The mesh was made using four-nodal quadratic dominated elements (CPE4RT) and had reduced integration under plane strain assumption, which provided an accurate stress distribution [21]. The TGO layer was discretized by vectorization according to the real structures of TBCs, and the size of smallest element was in the order of 1 μm.

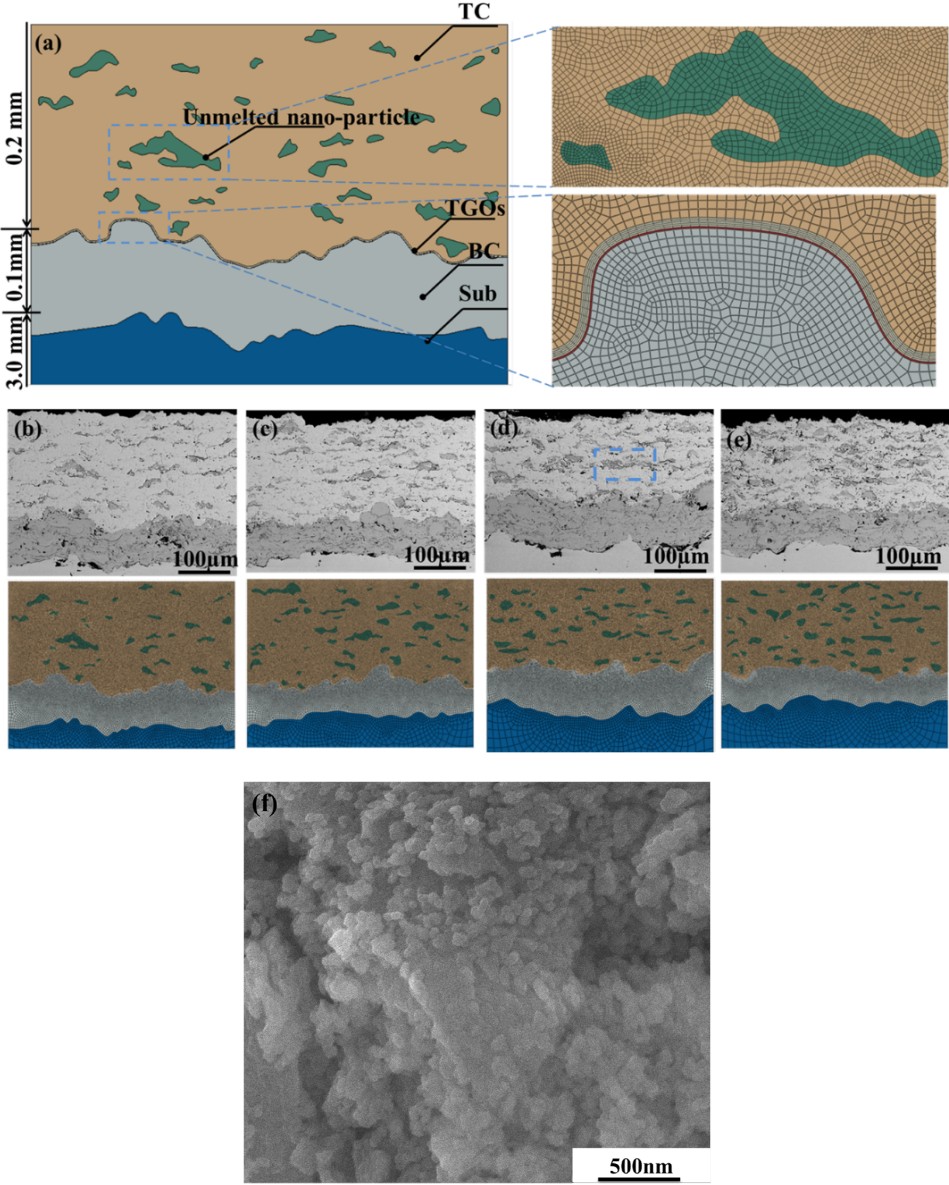

**Figure 1.** (**a**) Finite element model of thermal barrier coatings (TBCs) system and a detailed finite element mesh in the top coat (TC) and thermally grown oxide (TGO)/TC interface; (**b**) cross-sectional SEM image and corresponding model of NC1 with 11% unmelted nano-particle content (UNCs); (**c**) cross-sectional SEM image and corresponding model of NC2 with 15% UNCs; (**d**) cross-sectional SEM image and corresponding model of NC3 with 22% UNCs; (**e**) cross-sectional SEM image and corresponding model of NC4 with 34% UNCs, (**f**) the detailed morphology of unmelted nano-particles in NC3.

### 2.2.2. Material Properties

All the layers are considered to be isotropic and homogenous. The TC is treated as an elastic material, whereas TGO, BC and substrate are considered as an elastoplastic material. The properties of materials are temperature dependent, as listed in Tables 1–3 [22–30]. The influence of UNPs on the heat transfer and stress distribution of TBCs system is primarily introduced. The thermal conductivity calculation is based on the classic Fournier heat transfer law.

**Table 1.** Material thermophysical parameters of TBCs system.

| Materials | $T/^{\circ}C$ | $E/GPa$ | $\alpha/10^{-6} \cdot K^{-1}$ | $\nu$ | $k$ /$W \cdot m^{-1} \cdot K^{-1}$ | $C$ /$J \cdot kg^{-1} \cdot K^{-1}$ | $\varrho/kg \cdot m^{-3}$ |
|---|---|---|---|---|---|---|---|
| | 25 | 48 | 7.9 | 0.25 | 1.5 | 500 | 5280 |
| | 200 | 47 | 8.7 | 0.25 | 1.2 | 535 | 5280 |
| YSZ | 400 | 43 | 9.4 | 0.25 | 1.2 | 576 | 5280 |
| | 800 | 39 | 16 | 0.25 | 1.2 | 637 | 5280 |
| | 1100 | 25 | 16 | 0.25 | 1.1 | 637 | 5280 |
| Unmelted nano-particle | 25 | 10 | 7.9 | 0.25 | 0.5 | 300 | 3580 |
| | 25 | 152 | 12.3 | 0.3 | 4.3 | 501 | 7320 |
| | 200 | 143 | 13.2 | 0.31 | 5.2 | 546 | 7320 |
| BC | 400 | 133 | 15.2 | 0.31 | 6.4 | 592 | 7320 |
| | 800 | 118 | 16.3 | 0.32 | 10.2 | 781 | 7320 |
| | 1000 | 74 | 17.2 | 0.33 | 16.5 | 781 | 7320 |
| | 1100 | 41 | 17.7 | 0.33 | - | 781 | 7320 |
| | 25 | 400 | 7.1 | 0.27 | 5.8 | 600 | 4200 |
| | 200 | 390 | 7.5 | 0.27 | 5.8 | 600 | 4200 |
| TGOs | 400 | - | - | 0.27 | 5.8 | 600 | 4200 |
| | 800 | 355 | 9.0 | 0.27 | 5.8 | 600 | 4200 |
| | 1000 | 325 | 9.5 | 0.27 | 5.8 | 600 | 4200 |
| | 1100 | 315 | 9.7 | 0.27 | 5.8 | 600 | 4200 |
| | 25 | 204 | 12.6 | 0.32 | 11.5 | 431 | 8110 |
| | 200 | 195 | 14 | 0.32 | 14.6 | 465 | 8110 |
| Sub | 400 | 179 | 14.4 | 0.33 | 17.5 | 494 | 8110 |
| | 800 | 149 | 15.4 | 0.34 | 23.8 | 682 | 8110 |
| | 1000 | 137 | 16.3 | 0.34 | 33.1 | 833 | 8110 |

**Table 2.** Plastic parameters of the base coat (BC).

| $T/^{\circ}C$ | Stress/MPa | Plastic Strain |
|---|---|---|
| 25 | 1000 | 0 |
| 400 | 2500 | 0.23 |
| 600 | 2200 | 0.30 |
| 800 | 375 | 0.02 |
| 900 | 60 | 0.02 |
| 1000 | 19 | 0.01 |

**Table 3.** Creep parameters of TBCs.

| | $B/s^{-1} MPa^{-n}$ | $n$ | $T/^{\circ}C$ |
|---|---|---|---|
| TC | $1.8 \times 10^{-10}$ | 1 | 1000 |
| TGOs | $7.3 \times 10^{-8}$ | 1 | 1000 |
| BC | $6.5 \times 10^{-19}$ | 4.6 | $\leq 600$ |
| BC | $2.2 \times 10^{-12}$ | 3.0 | 700 |
| BC | $1.8 \times 10^{-7}$ | 1.6 | $\geq 800$ |

2.2.3. Thermal Loads and Cracking Behavior

The thermal loads consist of three steps, as shown in Figure 2. Firstly, the coatings (NC1–NC4) were heated from 25 to 1250 °C in 300 s; then a holding-time at 1000 °C for 3600 s; and finally, a cooling step from 1250 to 25 °C in 300 s. The temperature parameters of the bottom of substrate during the thermal shock resistance test are listed in Table 4. The linear growth of TGO layer is introduced in the model. All layers in TBCs system are assumed as homogeneous and isotropic. The crack propagation in TC is assumed to be mixed mode. The crack growth is controlled by a power law:

$$\left(\frac{G_n}{G_{nc}}\right)^{\alpha_m} + \left(\frac{G_s}{G_{sc}}\right)^{\alpha_n} = 1 \tag{1}$$

where $G_n$ and $G_s$ are the critical strain energy release rate (SERR) in modes I and II, respectively [31]. The predefined cracks will propagate when $\alpha_m$ and $\alpha_n$ reaches 1.0. The tensile strength of the TC is set as 200 MPa [32]. The critical maximum principal stress is considered to be 15 MPa and the critical fracture energy release rate is set as 15 J·m$^{-2}$ [28,30]. The displacement vector function $u$ is as follows:

$$u = \sum_{I \in N} N_I(x) \left[ u_I + H(x)a_I + \sum_{a=1}^{4} F_\alpha(x)b_I^\alpha \right] \tag{2}$$

where $N_I(x)$ is conventional shape function; $N_I$ is the usual nodal displacement vector related to the continuous part of $F_\alpha$ solution; $\alpha$ is the products of the nodal enriched degrees of freedom (DOFs); $H(x)$ is the discontinuous jump function; $F_\alpha(x)$ is the crack tip asymptotic function. More details are in reference [33]. The model with a detailed predefined crack setting is shown in Figure 3. For simplifying the analysis process, a circular particle with diameter of 10 μm was introduced into the model. The cracks in different directions were marked as shown in Figure 3a. The two cracks parallel to the *x*-axis are marked as *a* and *b*; the four cracks 45° from the *x*-axis are named *c–f*; and two cracks parallel to the *y*-axis are labeled A and B. According to the distribution of UNPs in TBCs, one particle, three particles and five particles were set respectively in the XFEM models, as shown in Figure 3b.

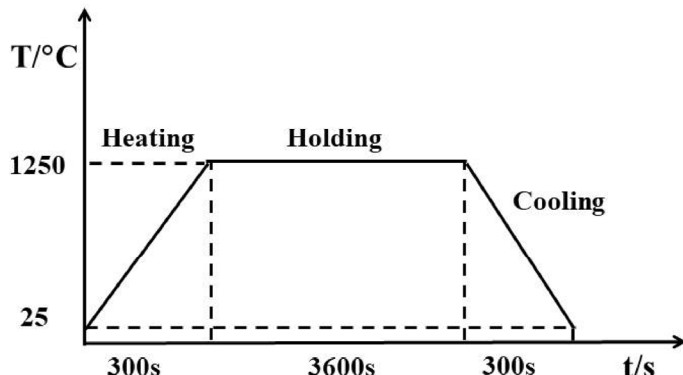

**Figure 2.** Thermal loading history for the simulation model including three steps.

**Table 4.** Temperature parameters of thermal shock test at the bottom of substrate.

| Samples | T/°C |
|---------|------|
| NC1 | 1115 |
| NC2 | 1110 |
| NC3 | 1090 |
| NC4 | 1080 |

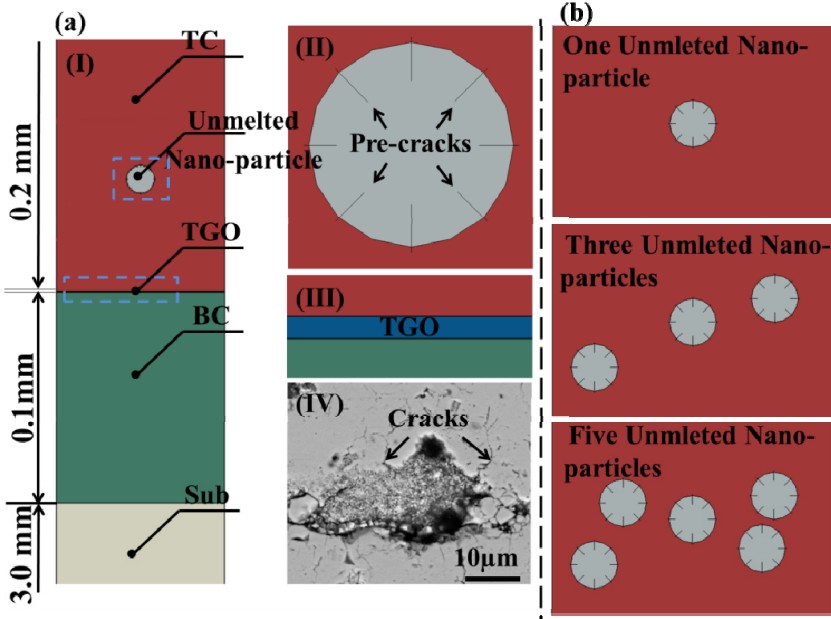

**Figure 3.** Crack propagation model of TBCs. (**a**) All the layers of TBCs; a detailed view of UNP with predefined cracks; TGO layer; and an SEM image of UNP with microcracks around it; (**b**) various UNCs with predefined cracks.

## 3. Results and Discussion

### 3.1. Analysis of Stress Field Distribution

The Mises, normal $\sigma_{22}$ and shear stress $\sigma_{12}$ distribution of TC in NC1–NC4 coatings are in Figure 4. From Figure 4, all the types of stresses are highly dependent on the surface roughness of BC/TC interface or UNP/crystalline region's interface. For the mises stress in Figure 4a, due to the loose distribution structure, low elastic modulus and low density, thermal stress inside the UNPs presented the lowest value, thermal stress inside the UNPs presented the lowest value (blue area, 150–317 MPa) and the stress in the vicinity of the unmelted nano-particle was higher (green area, 317–483 MPa), whereas the stress in the crystalline regions was the highest (yellow and red areas, >483 MPa). The above results suggested that the increase of unmelted nano-particle content (UNCs) played a decisive role in reducing the overall thermal stress. The tangential stress $\sigma$ in plane can be expressed thusly [34]:

$$\sigma = \frac{E_{TBC}}{1 - \nu_{TBC}}(\alpha_{TBC} - \alpha_{\text{sub}})\Delta T \tag{3}$$

where $E_{TBC}$ is the effective elastic modulus of TC; $\nu_{TBC}$ is poison's ratio; $\alpha_{TBC}$ and $\alpha_{\text{sub}}$ are the thermal expansion coefficients of both TC and the substrate, respectively. With the increase of UNC, the compressive stress areas of TC increased, while the elastic modulus and thermal stress of TC decreased. It is well known that the $\sigma_{22}$ and $\sigma_{12}$ tend to cause mode I and mode II fractures, respectively. Additionally, the normal and shear stress were considered as the main driving force of crack propagation. As for the normal stress $\sigma_{22}$ in Figure 4b, the tensile stress existed in the yellow areas and the blue areas corresponded to the compressive stress. Due to the thermal expansion coefficient of TC being higher than that of TGO, the contraction rate of former was higher than for the latter. The maximum tensile stress is at the peak of the TC/TGO interface (>50 MPa), while the compressive stress is at the valley. For a single unmelted nano-particle, the tensile stress and compressive stress existed at the peaks of convexity and concavity, respectively. Meanwhile, the tensile stress almost generated symmetrically outside the UNPs, which easily resulted in the initiation and propagation of microcracks at the interface between the UNPs and the crystalline regions.

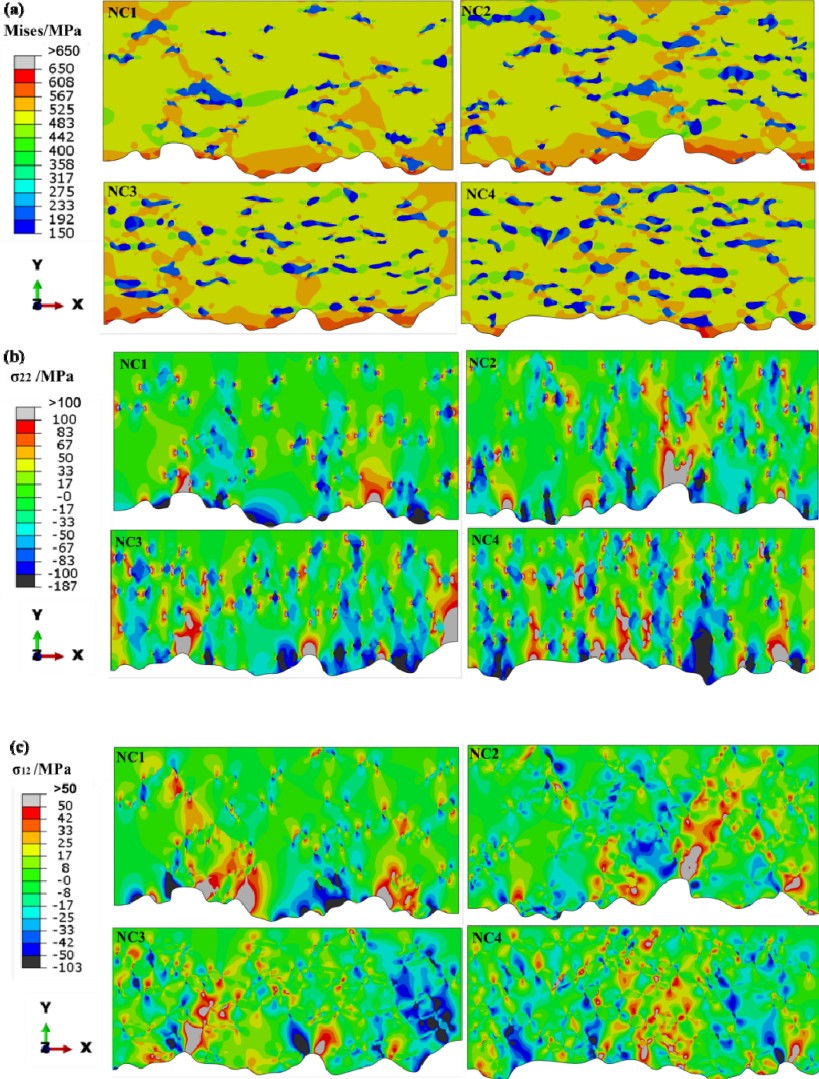

**Figure 4.** (**a**) Misses stress; (**b**) $\sigma_{22}$ and (**c**) $\sigma_{12}$ stress distribution of top coats in NC1, NC2, NC3 and NC4.

To further characterize the role of UNPs in the stress distribution in TC, the tensile stress $\sigma_{22}$ in NC4 along the predefined path with white line was extracted as shown in Figure 5. The positive "+" and negative "−" signs are used to visually represent the tensile and compressive stresses (see Figure 5a), respectively. There were many UNPs continually appearing along the white line, and the tensile and compressive stresses were gradually alternated. In addition, the value of stress significantly increased around the UNPs (see Figure 5b). In the crystalline region far away from the UNPs, the value of tensile stress in the crystalline region nearly stayed at 0–33 MPa, which was even lower than that of area around the UNPs, indicating that the UNPs can reduce the overall stress of TC.

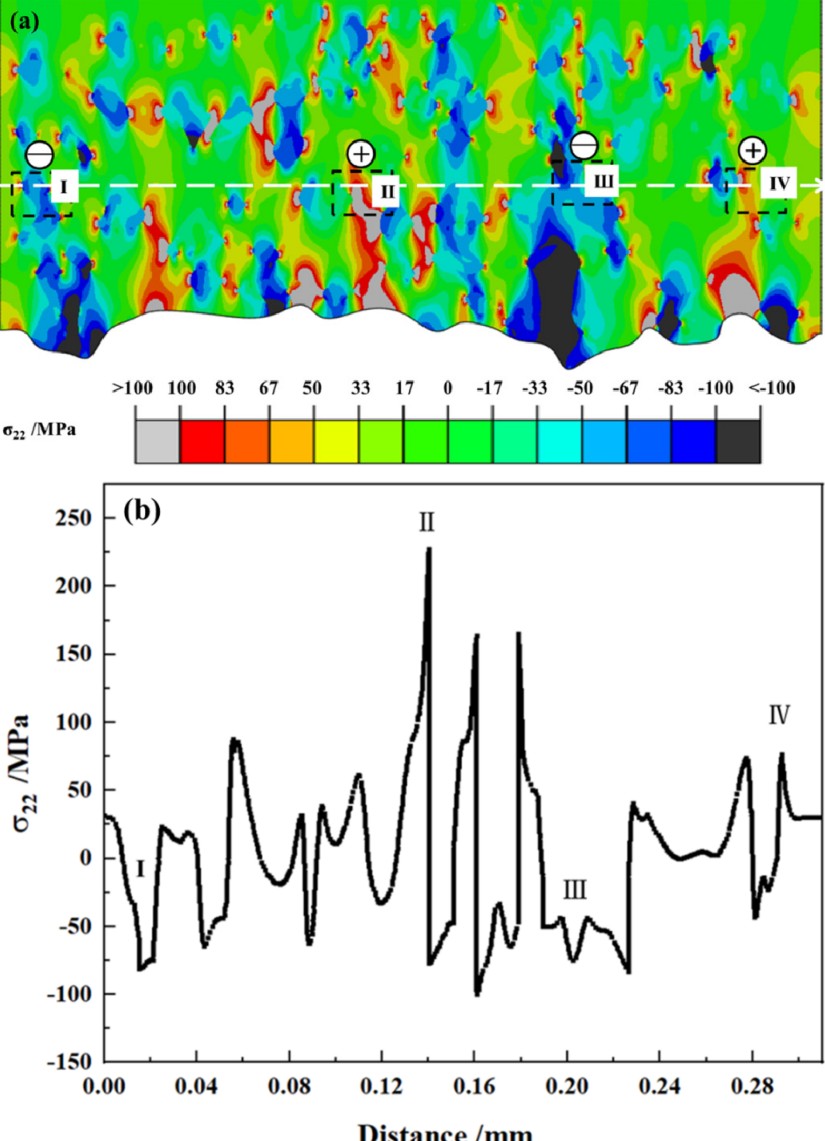

**Figure 5.** $\sigma_{22}$ stress distribution in the top coat (**a**) along the specified path; (**b**) corresponding stress distribution curve.

Figure 6 gives the $\sigma_{22}$ and $\sigma_{12}$ stress distribution in BC. As shown in Figure 6, since the tensile stress or shear stress reached the maximum in the BC, the initiation and propagation of cracks were both prone to occurring in the vicinity of TC/TGO or TGO/BC interface, which was consistent with the previous experiment and simulation results [19,30]. The values of tensile or shear stress in the TC were relatively lower than that of BC, indicating that the addition of UNPs was helpful for relieving the thermal stress.

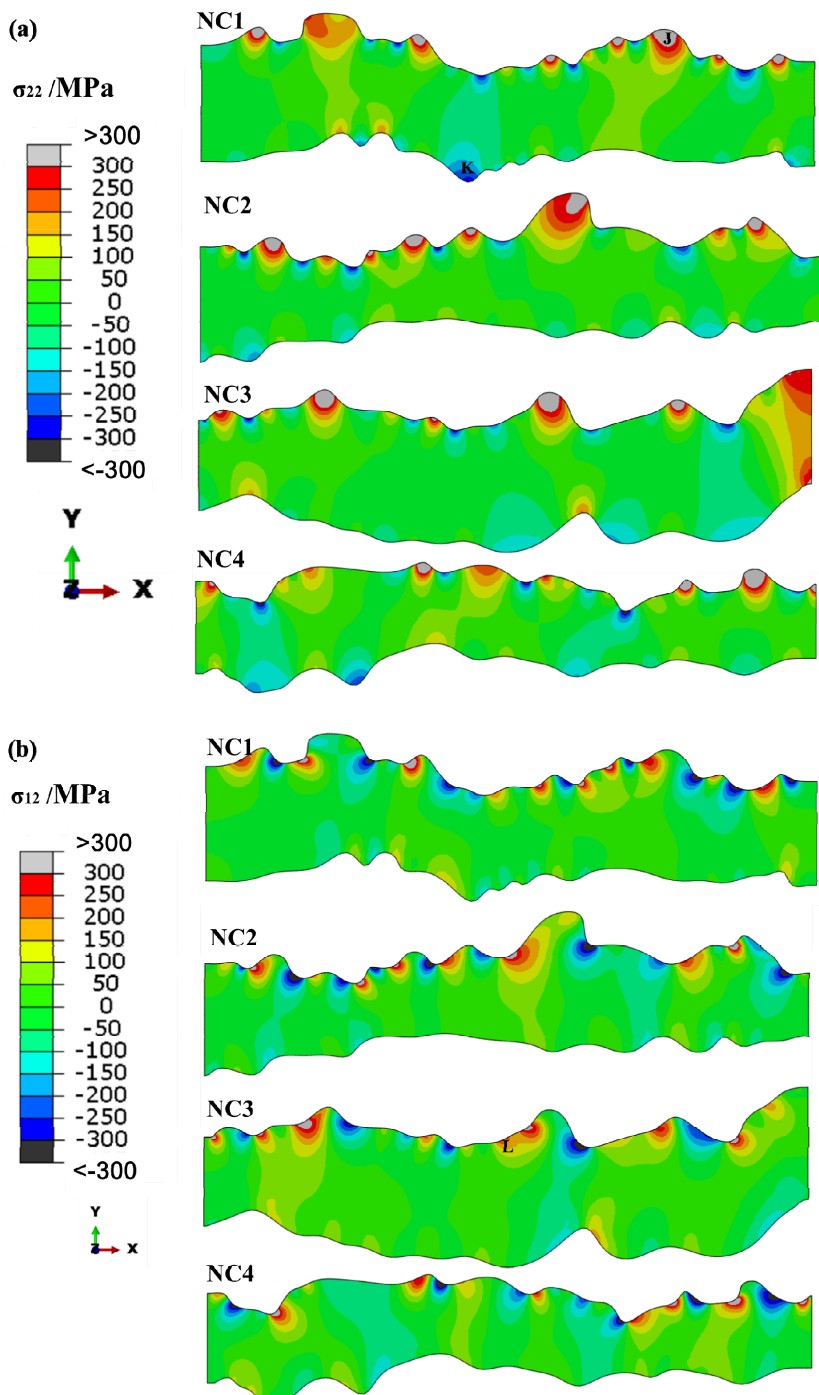

**Figure 6.** (**a**) $\sigma_{22}$ and (**b**) $\sigma_{12}$ stress distribution in BC.

Figure 7 depicts the $\sigma_{22}$ and $\sigma_{12}$ stress distribution in the TGO layer. As seen from Figure 7, the maximum stress of the whole TBCs concentrated in the TGO layer. During the thermal cycling, the thermal mismatch stress formed in the vicinity of TGO layer due to the differences in the thermo-mechanical properties of each layer. The tensile or shear stress of TGO layer was approximately 2–4 GPa, which is obviously larger than the value of TC or BC. The large tensile or shear stress easily resulted in the premature failure of TBCs at the TC/TGO or TGO/BC interface.



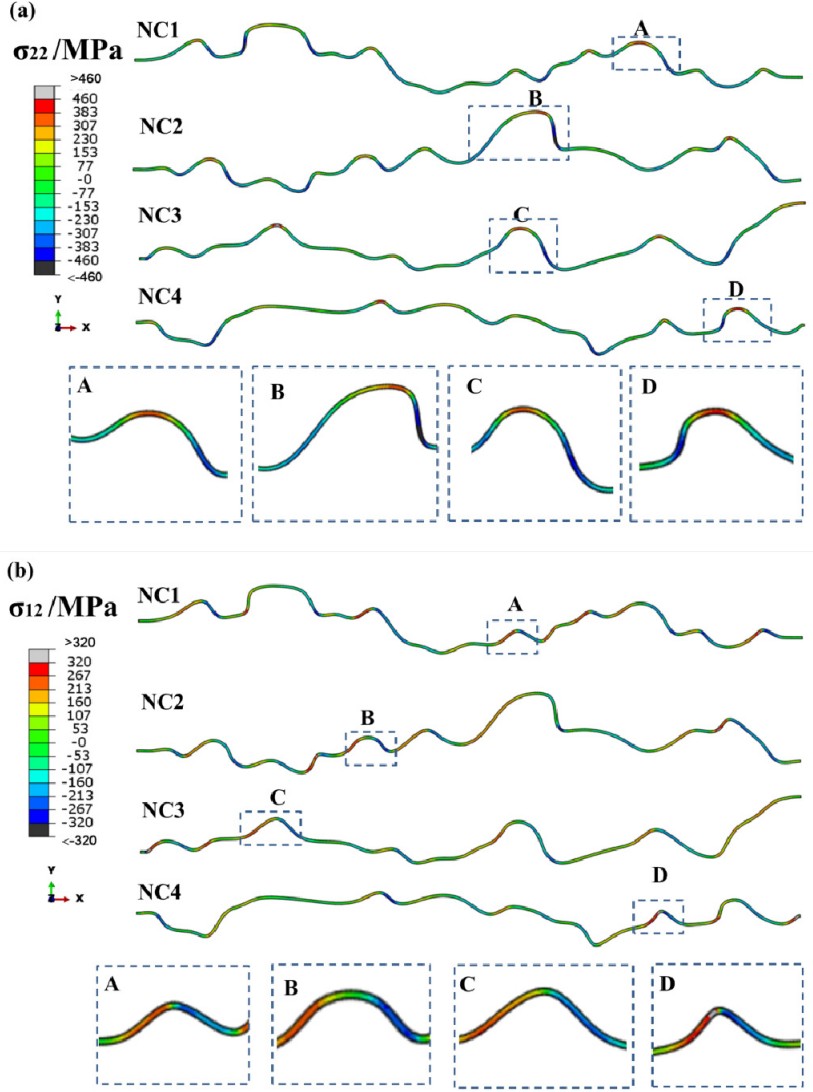

**Figure 7.** (**a**) $\sigma_{22}$ and (**b**) $\sigma_{12}$ stress distribution in TGO layer.

### 3.2. Effects of Unmelted Nano-Particles and Their Content on the Cracking Propagation Behavior

To better understand the failure modes and processes of nanostructured TBCs, some cracks in various orientations, such as parallel, perpendicular and 45° to the *x*-axis, were predefined at the UNPs/crystalline region interface, as shown in Figure 3a. The XFEM method was employed to stimulate the propagation paths of cracks. Figures 8 and 9 showed the time-dependent crack propagation and stress distribution in TBCs with various UNCs during thermal cycling. As seen from Figures 8a and 9a, there was only one unmelted nano-particle per unit area system, corresponding to the condition of low UNCs (NC1 and NC2). At the early stage, two horizontal cracks propagated parallel to the *x*-axis and then the propagation directions were mainly kept parallel to the *x*-axis under the tensile stress $\sigma_{22}$. The cracking due to stress-induced was prevented by the presence of compressive stress at the inner of the UNPs. The four cracks 45° from *x/y*-axis gradually propagated parallel to the *x*-axis due to the increased tensile stress $\sigma_{22}$ from 50 s to 150 s, indicating that the time-dependent crack propagation path was mainly determined by the tensile stress $\sigma_{22}$. Figures 8b and 9b showed three UNPs per unit area, which simulated the condition with medium UNCs like NC3. Six predefined cracks, including two cracks parallel to the *x*-axis and the other four 45° from the *x/y*-axis, propagated under the stress $\sigma_{22}$ and $\sigma_{12}$. As seen from Figures 8b and 9b, two horizontal cracks propagated parallel to the x-axis at the early stage and then the propagation direction kept constant under the tensile stress. Four cracks

with 45° from *x/y*-axis propagated slowly under the shear stress $\sigma_{12}$. However, in the final stage their propagation direction turned parallel to the *x*-axis due to the expanded area of tensile stress with the increase of thermal cycle. Figures 8c and 9c show five UNPs per unit area, corresponding to the condition with the high content of unmelted nano-particles such as NC4. The propagation tendencies of six cracks including those parallel to the x-axis and 45° from the *x/y*-axis were similar to the condition with one unmelted nano-particle. The tensile or shear stress areas were around the unmelted nano-particle I in the initial stage because a complex thermal stress area was formed by the accumulation of nano-particles [17,35]. In the final stage, both the tensile stress and shear stress coexisted inside the UNPs. Moreover, the length of the cracks with 45° from *x/y*-axis in the unmelted nano-particle was smaller, suggesting that firstly the mode II dominated the crack growth mode and it shifted to mixed mode until the crack propagated parallel to the *x*-axis. The influence of the shear stress on the longitudinal crack propagation behavior became more and more significant. Therefore, it can be predicted that the micro-cracks would enter into the nano-particles and were finally inclined to form some horizontal cracks, resulting in the spallation of TBCs.

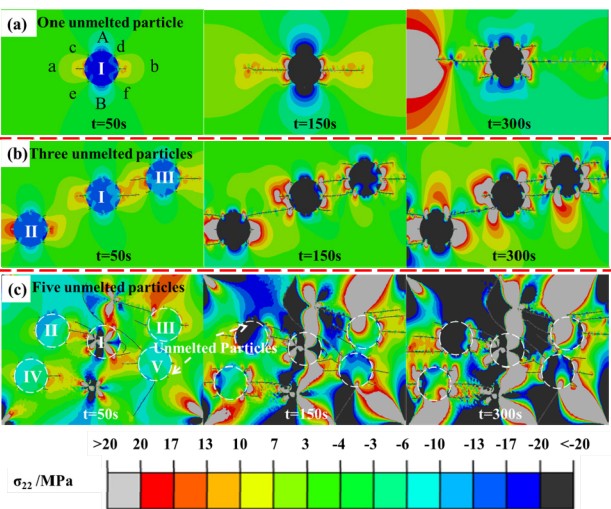

**Figure 8.** Crack propagation and $\sigma_{22}$ stress distribution in coatings with various UNCs, (**a**) one UNPs; (**b**) three UNPs; (**c**) five UNPs.

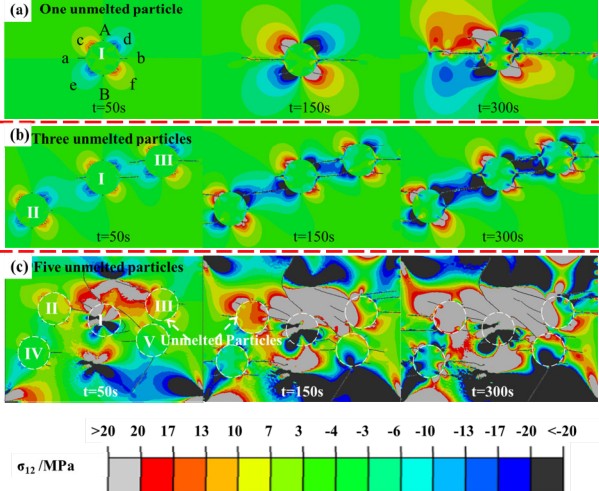

**Figure 9.** Crack propagation state and $\sigma_{12}$ stress distribution in coatings with various UNCs, (**a**) one UNPs; (**b**) three UNPs; (**c**) five UNPs.

Figure 10 described the crack propagation path in TBCs with various UNCs. The red and blue colors in Figure 10 show the initiation and propagation of cracks respectively. It can be seen from Figure 10 that the propagation direction of cracks was mainly along the horizontal direction and the obvious "capture effect" of UNPs on the surrounding cracks. This effect was enhanced with the UNCs increasing. The propagation crack angle is as follows [36]:

$$\theta = 2 \tan^{-1} \frac{1}{4} \left[ \frac{K_I}{K_{II}} \pm \sqrt{\left(\frac{K_I}{K_{II}}\right)^2 + 8} \right] \tag{4}$$

where $\theta$ is the crack propagation angle; $K_I$ is tensile stress intensity factor in fracture mode I; $K_{II}$ is tensile stress intensity factor in mode II. The maximum circumferential tensile stress intensity factor ($K_{max}$) can be calculated as:

$$K_{max} = \cos \frac{\theta}{2} \left( K_I \cos^2 \frac{\theta}{2} - 1.5 K_{II} \sin \theta \right) \tag{5}$$

where $K_{max}$ is the maximum circumferential tensile stress intensity factor. The propagation of the horizontal and longitudinal cracks was viewed as the spring vibrator [36]. When the distance between two adjacent UNPs was small enough, the stress field would be overlapped, leading to the further amplification for the crack propagation. In addition, under the interaction of tensile-tensile stress field, the propagation of the crack and its length can be further promoted, while under the tensile-compressive or compressive-compressive stress field interaction, the propagation of the crack can be prevented.

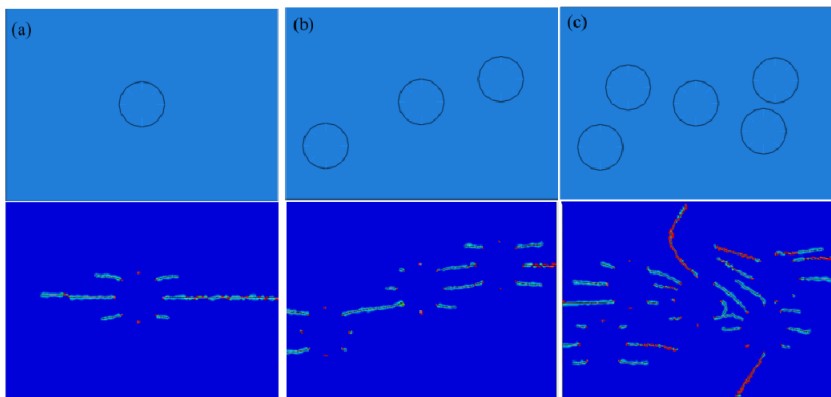

**Figure 10.** Crack propagation path in the coatings with various UNCs, (**a**) a single UNP; (**b**) three UNPs; (**c**) five UNPs.

Figure 11 shows the experimental results from the thermal cycling test. As can be seen from Figure 11a,b, the crack propagated mainly along the horizontal direction at the UNPs/crystalline region interface. When the UNCs is relatively low, it is easy to form some penetrating horizontal cracks. However, these cracks cannot enter into the UNPs, which can effectively prevent the propagation of cracks (see Figure 11c). With the increase of UNCs, the UNPs show an obvious "capture effect" on the surrounding cracks (see Figure 11d). The experimental results are in a good keeping with the simulation ones. Figure 12 gives the schematic diagram of the "capture effect" of UNPs. The UNPs and surrounding microcracks had a mutual interaction effect on the stress concentration and crack propagation. As shown in Figure 12a, due to the compressive stress generated inside or around the UNPs, some cracks can only propagate towards the tensile stress zone of the adjacent UNPs. When the unmelted nano-particle was gradually subjected to sintering and the nanostructure gradually disappeared, the crack could enter into the unmelted nano-particle accompanied by the change of propagation direction (see Figures 11d and 12b).

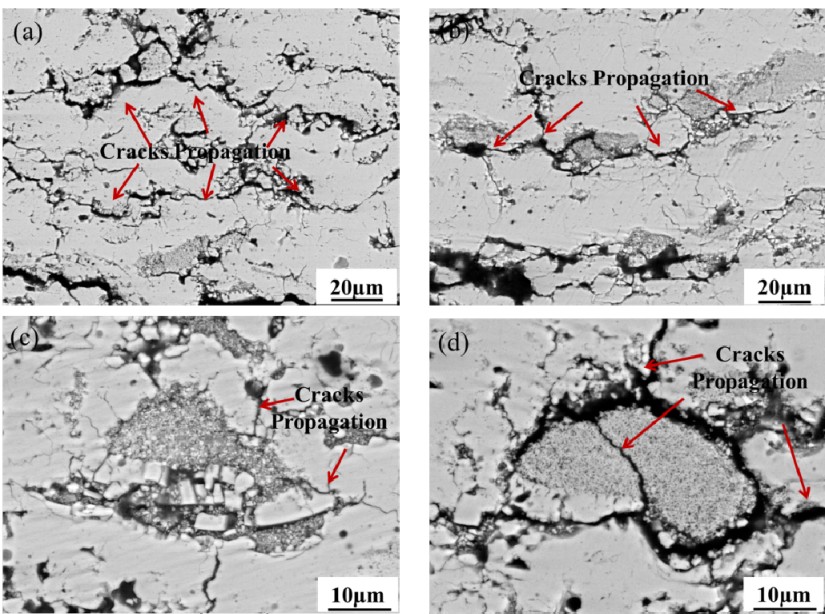

**Figure 11.** Effect of UNPs on cracks propagation behavior. (**a**) Horizontal propagation; (**b**) capture effect; (**c**) inhibition effect of cracks propagation by interface; (**d**) change of propagation direction.

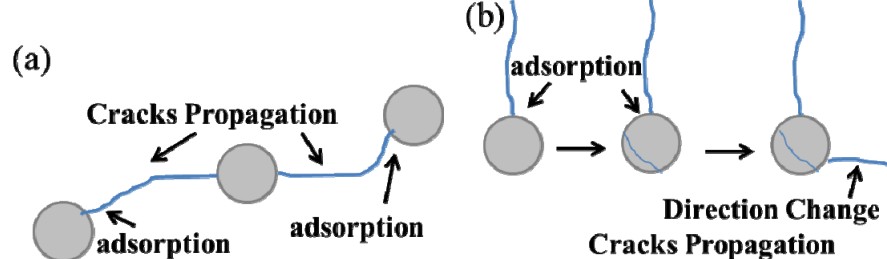

**Figure 12.** Schematic diagram of inhibiting effect of cracks propagation from UNPs: (**a**) cracks were stopped at the UNPs and crystalline region interface; (**b**) cracks entered the UNPs.

Based on the above results in the simulation and experiments, the failure mechanisms of nanostructured TBCs can be summarized as follows:

(1) Accumulation of thermal stress: The tensile stress is mainly distributed in the horizontal or vertical direction of the UNPs. The compressive stress distributes inside the UNPs and the shear stress presents symmetrical distribution around the UNPs.

(2) Propagation of horizontal cracks: Under the tensile and shear stress, the cracks mainly propagated along the horizontal direction. The predefined cracks with 45° from the *x*/*y*-axis were a type I and II mixed-mode cracks. These cracks propagated along the direction parallel to the *x*-axis since the tensile stress $\sigma_{22}$ and the shear stress $\sigma_{12}$ were the main driving forces for the cracks propagation.

(3) "Capture effect" of UNPs: Cracks tended to propagate towards the tensile stress region of the surrounding UNPs. When the crack entered into the low elastic modulus and loose porous UNPs, the crack propagation was prevented.

(4) Experimental observation for spallation of TC: With the thermal cycles increasing, the ability of UNPs to prevent crack propagation decreased and the crack eventually entered into the UNPs or propagated along the interface between the UNPs and crystalline regions, resulting in the spallation of TC (Figure 11d).

The change of crack path was attributed to the reduced stress concentration of the crack tip and compressive stress inside the UNPs. Therefore, the UNCs played a significant role in prolong of lifetime

of nanostructured TBCs during thermal cycling. The reason was that these UNPs seriously affected the value and distribution of residual stress. The increase of UNCs effectively decreased elastic modulus and residual stress of TC. However, once the UNCs was less than a certain range, the mechanical properties of TC decreased significantly, and further shortened the lifetime of the TBCs during thermal cycling. Combined with the simulation and previous experimental results above, it was found that when the UNCs was in a range of 20–30%, the TBCs had superior thermal shock resistance.

## 4. Conclusions

In this study, an extended finite element model (XFEM) based on the real microstructure of nanostructured TBCs was built and employed to elucidate the correlation between the microstructure and crack propagation behavior. The effect of content of unmelted nano-particles on the cracking initiation and propagation behavior were obtained. Some important conclusions are as follows:

(1) During the thermal cycling, the UNPs can effectively reduce the thermal stress of TC. The tensile stress and shear stress regions outside the UNPs enhance the initiation of cracks, while the compressive stress inside the UNPs can effectively prevent the cracks propagation.

(2) Arbitrarily oriented cracks mainly propagated parallel to the x-axis at the final stage of thermal cycle, indicating that tensile stress was the main driving force for the spallation failure of TBCs. Correspondingly, I and I–II mixed types of cracks are the major cracking failure patterns.

(3) The UNPs that distributed in the nanostructured coating had an obvious "capture effect" on the cracks, which means that many cracks easily accumulated in the tensile stress zone of the adjacent UNPs and a complex microcrack network generated at the periphery of UNPs.

(4) At the final stage of thermal cycling, the cracks eventually entered into the UNPs or propagated along the interface between the UNPs and crystalline region. Both the tensile stress and shear stress of TC were lower than those of BC. The spallation failure usually occurred at the TC/TGO interface.

**Author Contributions:** Conceptualization, L.Z. and Y.W.; methodology, L.Z.; software, L.Z.; validation, Y.B. and Y.W.; formal analysis, L.Z. and W.F.; investigation, L.Z. and Y.G.; resources, L.Z.; data curation, Y.G. and Y.S.; writing—original draft preparation, L.Z.; writing—review and editing, L.Z., Y.W., and Y.B.; visualization, Y.B. and Y.W.; supervision, Y.W., W.F., Y.G. and Y.S.; funding acquisition, Y.W. and Y.B. All authors have read and agreed to the published version of the manuscript.

**Funding:** This work was supported by National Key R&D Program of China (grant number 2018YFB2004002), the China Postdoctoral Science Foundation (grant number 2019M653598), the State Key Laboratory of Electrical Insulation and Power Equipment (grant number EIPE20301) and the Natural Science Foundation of Shaanxi Province (grant numbers 2019TD-020, 2019JQ-586 and 2020JQ-911).

**Acknowledgments:** Thanks Zhiyuan Wei for the technical support and guidance to the FEM simulation.

**Conflicts of Interest:** The authors declare no conflict of interest.

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
