# Peer review of "A Simulation Study on the Crack Propagation Behavior of Nanostructured Thermal Barrier Coatings with Tailored Microstructure"

_coatings, doi:10.3390/coatings10080722_

Round 1

Reviewer 1 Report

This is a well prepared and presented work.

Please try to stress out further the rationale and the merit of this work especially in comparison to other works.

I would personally appreciate more recent publications dealing with alternative TBC materials and or formulations. Therefore i suggest to keep review articles but maybe it would be good to update or replace some of the decade+ old publications. Several works have been published in MDPI journals that you may try such as “Coatings” “Metals” “Nanomaterials” etc.

Also other perovskite type materials has been studied as TBC component https://doi.org/10.1016/S1003-6326(18)64800-9 in either layered systems https://doi.org/10.1016/j.prostr.2018.09.039 or nanocomposites https://doi.org/10.1016/j.matchemphys.2020.123093

Since TGO is considered as 1um thick is the 1um element size of proper dimensions? Please comment

The thermal shock test conditions are not clear. Is the specimen placed in an oven? Is it cooled on one side or not? Is it at ambient air? Some experimental information should be reproduced here even if it is previously published.

Specimens’ size, shape, edges and substrate alloy type should be provided. Are the specimens coated on one side?

What is the reason for the difference in the back side temperature among NC1-NC4 samples?

The authors refer to (line 320)“at the final stage of thermal cycles, indicating that…” or (line 296) “TC: with the thermal cycles increased, the…”  It is not clear how many thermal shock cycles have been applied.

Quality of figures is good but in cases of figures with two or more parts it is better to be found in the same page (e.g Fig.5 / 6 /7).

Minor typos exist (e.g. line 284)

Reviewer 2 Report

The presented work is relevant. The influence of microstructure of thermal barrier coatings on the behavior of fatigue cracks was studied. It is noted that unmelted nano-particles in the coating microstructure have a “capture effect” on the cracks, which leads to a complex network of microcracks. It is noted that the tensile stresses in the material are the main driving factor in the failure of the thermal barrier coating. An extended finite element model is presented that allows simulating the propagation of cracks without remeshing. The results of experimental studies that are consistent with the results of modeling are presented.

There are some comments:

  1. It is not clear that in this study the authors mean by nano-particles in the coating microstructure. From figure 1 and figure 11, it is not at all obvious that the unmelted particles have nanoscale dimensions; structural groups with micrometer dimensions are demonstrated.
  2. The crack can be stopped at any particle interface in the coating structure, not necessarily nanoscale.
  3. Figure 4a shows the Mises stress distribution, with stress values exceeding 650 MPa at some interface boundaries. The question arises: does the presented picture of stress distribution correspond to the final stage of loading or does it illustrate the stresses remaining in the material after all external loads are removed?
  4. 4. It would be useful to provide information on which software product the authors performed finite element modeling and which plasticity model they used.

Reviewer 3 Report

Authors have attempted to show the effect of presence of unmelted nano-particles (UNPs) on crack propgation and failure of TBCs using exteneded finite element (XFEM) simulations. 

The manuscript is well written howevere I have a major objection with the assumptions that authros have made reagrding the size of the nano-particles. 

The simulation was done aussiming 10µm diamaeter UNP which is very large to be called as a nano-particle. 

Therefore, i suggest either authors should reallign the objective of the article or they shoulld do the simulations with particles with diameter of <100nm. 

Round 2

Reviewer 3 Report

Authors have addressed the comments and i have no further objections.